# Food Patterns among Chinese Immigrants Living in the South of Spain

**DOI:** 10.3390/nu13030766

**Published:** 2021-02-26

**Authors:** Bárbara Badanta, Rocío de Diego-Cordero, Lorena Tarriño-Concejero, Juan Vega-Escaño, María González-Cano-Caballero, María Ángeles García-Carpintero-Muñoz, Giancarlo Lucchetti, Sergio Barrientos-Trigo

**Affiliations:** 1Research Group PAIDI-CTS 1050 Complex Care, Chronicity and Health Outcomes, Faculty of Nursing, Physiotherapy and Podiatry, University of Seville, 41009 Seville, Spain; bbadanta@us.es (B.B.); ltarrino@us.es (L.T.-C.); agcarpin@us.es (M.Á.G.-C.-M.); sbarrientos@us.es (S.B.-T.); 2Research Group CTS 969 Innovation in HealthCare and Social Determinants of Health, Faculty of Nursing, Physiotherapy and Podiatry, University of Seville, 41009 Seville, Spain; 3Research Group CTS 1054 Interventions and Health Care, Red Cross, Spanish Red Cross Nursing School, University of Seville, 41009 Sevilla, Spain; juanvegadue@gmail.com; 4School of Medicine, Universidade Federal de Juiz de Fora, CEP: 36036-900 Juiz de Fora, Brazil; g.lucchetti@yahoo.com.br

**Keywords:** eating habits, emigrants or immigrants, food practices, nutrition, Chinese

## Abstract

This article aims to explore the food patterns of Chinese immigrants living in Spain and to understand the factors associated with these behaviors. This qualitative ethnographic study included 133 Chinese immigrants; using interviews with scripts were based on the National Health Survey of Spain. Three categories were defined: “Differences between Chinese food and Western food”; “Products and dishes consumed by Chinese immigrants” and “Modification of eating habits”. Participants found a remarkable difference between eastern (i.e., vegetables and rice) and western (i.e., dairy, high-fat, bread) foods. They also experienced a change in their eating habits, mostly related to acculturation and lack of time. Chinese immigrants have different eating patterns as compared to the native population in Spain and this information could help in future healthcare strategies. Knowing the food culture could infer aspects, such as insertion or integration, and promoting health and well-being.

## 1. Introduction

The World Health Organization (WHO) defines social determinants of health as the conditions in which people are born, grow, live, work, and age [1]. In this context, health patterns are important markers of health and can have an influence on the prevention of non-transmissible diseases [2,3,4].

Health patterns may change based on a wide array of factors, such as socioeconomic status, food insecurity, food deprivation, region of origin [5], religious beliefs, and lack of availability of traditional food [6], predominantly affecting certain populations, such as immigrants. 

Previous studies have already evidenced important differences in diet and body care between the immigrant and non-immigrant population [7,8], showing that overweight and obesity increase with length of residence in Western countries [9,10] and that unfavorable effects of Western acculturation on health behaviors could also contribute to an increased risk of chronic diseases following migration [11]. 

Given that immigration into Europe is expected to continue to play a critical role in population growth over the next decades, immigrants’ health should be a public health priority. Despite this fact, clinical information systems have few data related to the health patterns of that population [12]. Available information is even more limited for the population of Chinese origin. 

In Spain, although Chinese is the fourth largest nationality of origin (4.6%) [13], language barriers, intense working conditions, and the use of traditional medicine are related to their exclusion in most health surveys [14,15]. 

The Chinese culture has been characterized as food-centered, which emphasizes diet therapy and food modification to maintain one’s health, such as fostering healthy habits and maintaining a balance of “hot” and “cold” food consumption to prevent illness [16]. However, in the last decades, an increase in the prevalence of being overweight among Chinese adolescents has been observed, particularly in urban and rural-to-urban migrant Chinese children [17], probably due to eating pattern changes, and higher family income [18]. 

Due to the increase of Chinese immigration in Europe and Spain, and the lack of consistent studies assessing food patterns in this population, this study aims to understand the perceptions of food patterns among Chinese immigrants living in the southern region of Spain and some of the factors influencing these patterns. These findings could help in the development of future health strategies, and they may be considered by health professionals in their clinical practices.

## 2. Materials and Methods

### 2.1. Design

A qualitative ethnographic study was conducted in the south part of Spain. According to Roper and Shapira’s framework [19], the ethnographic approach considers the holistic context in which meaning is assigned to experiences, in order to understand why groups of people do what they do. This approach is characterized by (a) an interpretative paradigm, which emphasizes that there is no single reality; (b) the interplay between the individual (emic) and the societal (etic); and (c) interest in culture and people’s lives in the everyday contexts [20].

### 2.2. Data Collection

Data collection consisted of semi-structured interviews with Chinese immigrants and field notes. Interviews were carried out by the main researcher (B.B.) over six months in 2016/2017.

This study took place in Andalusia, the southernmost region of Spain and Europe, where, according to the National Statistics Institute, the Chinese population constituted 9.7% of the Chinese foreigners in Spain [21]

The focus was on participants’ shared behaviors and experiences; thus, the investigation worked under the assumption that they share cultural perspectives, even if they do not know each other. Participants were recruited through Chinese businesses (e.g., bazaars, restaurants, grocery stores, technology stores, fashion stores, wholesale businesses), and community institutions (e.g., Asian cultural centers, educational institutions, and health services). Purposive sampling was used to select the participants. In order to increase the number of participants, the “snowball sampling” procedure was also used.

As inclusion criteria, researchers included participants if they were adult Chinese immigrants who immigrated to Spain, and were able to communicate in Mandarin Chinese, English, or Spanish. The researchers selected young adult Chinese immigrants, taking into account that, according to data from the National Statistics Institute, the average age of the foreign population in Spain in 2016 was 32.09 years and the highest concentration of Chinese population was between 25 and 54 years [21].

The interviews were carried out face-to-face, in participants’ workplaces, while the Chinese worked and attended to customers, and lasted 15 to 30 min. The main researcher (B.B.) interviewed the Chinese immigrants, and audiotaped and transcribed verbatim. Data collection continued until saturation criteria. If the participant wanted to be interviewed at another moment of the day (for instance, outside his/her working time), the researcher came back at another previously scheduled moment in an attempt to increase the responsiveness of the sample.

In line with the ethnographic methodology [19], field notes were obtained and included information about witnessed events, verbatim verbal exchange from informal conversations, and the researcher’s personal interpretations of events. All of the information allowed the researcher to examine if interpretations of meanings behind observed behavior coincided with participants’ own understandings.

### 2.3. Measures

The interview script was based on the National Health Survey of Spain (NHSS, 2016/2017) [22]. This is a nation-wide study collecting health information (e.g., sociodemographic data, health status, and associated factors) on the population living in Spain. Participants answered brief quantitative questionnaire, and were submitted to a comprehensive qualitative interview using the following open questions: How is the daily diet of Chinese immigrants in Spain? What kind of products do they consume and where do they buy them? What breakfast, lunch, and dinner do they have? What are the main characteristics of Chinese food? Have you noticed any changes in the diet patterns after emigration to Spain? 

### 2.4. Data Analysis 

Quantitative data were analyzed using frequency, percentage, mean, standard deviations, chi-square (categorical data) and Mann–Whitney tests (ordinal or continuous data). The qualitative analysis was carried out following the steps proposed by Braun [23]: (1) familiarization with the data; (2) generation of categories; (3–5) search, review, and definition of themes; and (6) the final report. The data were obtained using a field diary and audio recording. 

Transcription, literal reading, and theoretical categorization were performed, and the NUDIST NVivo (version 12) software was used. Data analysis started with individual readings of all field notes and interview transcriptions, several times, to gain an overall understanding of the content. The analysis continued by organizing descriptive labels, focusing on persistent concepts and similarities/differences in participant behaviors and statements. In order to develop categories of meanings, the coded data from all of the participants were compared. A final report was prepared with the statements of the Chinese immigrants “C-questionnaire number, sex, age”. This research followed the criteria of the Consolidated Criteria for Reporting Qualitative Studies (COREQ) (Appendix A). 

### 2.5. Ethical Considerations

The study was approved by the blinded Ethics Committee (Code: blinded). All participants received information about the study, both oral and written. They knew their right to withdraw and the guarantee of anonymity. All participants included in the study signed the informed consent.

## 3. Results

A total of 252 businesses and institutions were visited and 133 Chinese immigrants agreed to participate. The sample consisted of 61.7% Chinese immigrant women and 38.3% men. All Chinese immigrants were 18 years or older and under 55, constituting a mean age of 30.7 years and with an average length of residence in Spain of 11.3 years (Table 1). The majority of Chinese immigrants were Han people, the dominant ethnic group of China, and they came from rural areas of China (Zhejiang) (78.3%). Recognizing limited possibilities of work in their place of origin, and having a medium–low educational level (71%), led them to emigrate to improve their family situations: (I-62, man, 32 years) “Most of the Chinese in Spain have a very low cultural level, so all they want to do here is work”. 

Regarding food patterns among Chinese immigrants, the most consumed foods (three or more times a week) were vegetables (94%), pasta, rice, or potatoes (87.2%), and meat (82.7%). Among the products least consumed (less than once a week, almost never or never), were chocolate (78.3%), butter (73%), cold cuts (67.2%) and bacon, and/or sausages (65%).

Immigrant businesses are places where the Chinese immigrant population spends most of their time. Given that employment is the main reason why they migrate to Spain, they take it as a way of life, so they want to always be working: (I-81, woman, 26 years) “If you don’t work, you are useless”. An example of this assumption is the fact that many businesses in Spain close for lunch, but not Chinese ones, which typically stay open all day. For this reason, most meals were taken at the workplace and were based on Chinese products and typical Chinese dishes as observed by the main researcher. Beyond the type of food, the long working hours were potential barriers for preventing a healthier diet and immigrants tended to eat in the intervals between clients.

The analysis of the structured interviews resulted in three categories: “Differences between Chinese food and Western food”; “Products and dishes consumed by Chinese immigrants” and “Modification of eating habits”.

### 3.1. Differences between Chinese Food and Western Food 

While comparing Chinese against the Spanish cuisine, immigrants have the general belief that “Chinese food is healthier than Western food”. However, different from Western persons, most of these immigrants usually do not categorize the food as “good” or “bad”. In fact, the principles of traditional Chinese medicine (TCM) are associated with the relationship between food and health, aiming to guarantee the balance between cold and heat, yin and yang. Thus, TCM has an influence on many dietary behaviors of this population: (C-115 man, 40 years) “We identify foods with elements of the TCM, classify them in different ways, and use them to restore internal balance”. The role of the TCM was observed in different aspects by the main researcher, ranging from the use and storage of medicinal herbs to books of TCM in the shelves of their businesses. A Chinese woman also commented that, during her pregnancy, she did not take yin foods (cold foods), such as bananas or cold drinks, as it would cause her to lose her balance, and affect the baby (C-131 woman, 27 years).

Other characteristics that differentiate the pattern of Chinese and Western Spanish food are the attributions of good luck for specific types of food (C-108 woman, 33 years “The fish is eaten a lot in the New Year, it brings good luck (Fish consumption in New Year means niannianyouyu: have abundance year-after-year (an auspicious saying for the lunar New Year)). Peaches for example, also bring good luck. In our culture you can see a lot of fruits with connotations”); the importance of food aesthetics (C-25 woman, 34 years) “In Chinese food the smell, taste and color are very important”; the variety of dishes (C-115 man, 40 years) “We usually have several small plates and they take all of them (…), we don’t eat a single meat dish or a steak like here”; and the way to prepare and cook food: (C-10 man, 32 years) “Many foods are made steamed (for example fish) or in wok. A very high temperature is achieved for a very short time, so vegetables and fish do not lose their properties”. The cultural beliefs associated with food favor the maintenance of this dietary pattern and the Chinese ethnic identity.

Despite the fact that Chinese immigrants usually maintain their Chinese culinary customs, there are remarkable differences according to their place of origin: spicy food is consumed by people from Beijing and other areas of northern China, bittersweet food in Shanghai, and salty food in southern China (Canton “Guangzhou”). However, outside their homes, several participants declared that Chinese food was not truly authentic in the Chinese restaurants located in Spain: (C-72 woman, 43 years) “In Spain the food is pseudo-Cantonese, it has evolved to the taste of the Western palate”.

Although Chinese immigrants are incorporating some western foods, such as toast, cereals, or coffee with milk for breakfast, all participants reported that they prefer their original Chinese food: (C-36 man, 37 years) “We can go to a Spanish restaurant one day and eat tapas, but not daily”.

### 3.2. Products and Dishes Consumed by Chinese Immigrants

Rice or noodles are used as the main sources of carbohydrates and constitute the basis of Chinese food, being usually consumed in all meals (breakfast, lunch, and dinner). In addition, the consumption of vegetables, varied meats, eggs, and fish also stand out. In the fieldwork, the main researcher found that, if some food was left over in the lunch, the immigrants tended to use it in other meals, such as dinner or even breakfast. 

Unlike the large consumption of cow’s milk in Spain, Chinese immigrants maintain their habit of consuming soymilk. Some foods, such as dairy products, are avoided, due to gastric discomfort and gastrointestinal problems associated with its use: (C-119 woman, 29 years) “The gases caused by cow’s milk, is common in almost all Chinese, that’s why we drink our milk”. In the same way, the statements show a low consumption of high-fat foods to avoid heavy digestions: (C-69 woman, 38 years) “The Chinese love fresh food … so we do not like sausages, pizzas, or frozen food”. However, it was possible to note that the avoidance of milk was more common among older persons. 

There are some typical dishes consumed at any time of the year that were commonly cited by Chinese immigrants, such as rice balls and fermented flour dumplings (typical everyday food stuffed with meat, vegetables, or mushrooms, and steamed), huo guo, sweet or salty zongzi, and chop suey. Other dishes were associated with certain seasons of the year and festivities, such as tāngyuán, qingmingguo, and yuebing (Table 2). In the researcher’s observation, it was possible to see a wide range of Chinese dishes being consumed and these dishes were more common than the Western dishes, which were seldom consumed by the immigrants. 

### 3.3. Modification of Food Patterns

All immigrants say the availability of food does not constitute a barrier that affects their eating patterns. They noted that some products could be purchased in national supermarkets, and the exclusive raw materials of Chinese food are available in specialized supermarkets distributed in Spanish cities, some of them visited by the main researcher. 

Although entrepreneurs have more free time and can sometimes eat at home with the family, most Chinese immigrants usually eat at work: (I-130 man, 53 years) “My father had a restaurant, so the restaurant’s kitchen was the living room of my house. We only went to sleep at my house”. Almost all respondents report having workdays of more than 10 h, continuous (I-72, woman, 43 years) “many hours of daily work without rest or closure during the half day and including full weekends”. This would explain why some people bring prepared food from home and the existence of kitchens in the back of the businesses, as observed by the main researcher, where Chinese immigrants eat quickly and continue the workday. This also means that sometimes they skip a meal (C-63 woman, 22 years “We sometimes do not have breakfast because we go to the store quickly”), not having a pre-established meal schedule, and not dedicating sufficient time to it. 

In spite of this lack of time, women and younger persons tend to be more careful with their health and their eating habits. For example, women reported eating more fruit than men, something that is statistically significant in our findings (*p* = 0.013); 76.8% of women consumed fruit daily as compared to 54% of men. From all participants, at least 12% reported doing some type of diet in order to reduce their weight, 7.1% in order to treat health problems, and 3% in order to maintain their health status. Regarding sex, women performed more diets to lose weight (18.7%) than men (2%); it was statistically significant (*p* = 0.005).

This diet does not mean a change towards western food culture, but an adaptation of their food consumption aiming to achieve their objectives. Participants reported they reduced some foods not recommended in people with high blood pressure or high cholesterol, as well hyperproteic diets that accompanied intense physical activity in gyms. 

Nevertheless, the integration of unhealthy eating habits due to the acculturation process is observed among the younger Chinese people; they consume fats—including oils—more frequently (U = 1233 *p* = 0.012), with an average age of most consumers being 28.4 years (SD = 6.6) and of the least consumers (sometimes or never) being 32.1 years (SD = 8).

## 4. Discussion

This study further understood the food patterns among Chinese immigrants in Spain, revealing that there are potential gaps that need to be addressed by the Spanish public health. 

According to the results, there is a remarkable difference between Eastern and Western foods. For example, spicy food is consumed by most of the population in China, especially in the northern region [29,30,31]. There are also differences noted on the origin of food (predominantly based on plants in China as compared to meat and dairy products in Western countries), on the patterns of eating [32], and the strong relationship between food and traditional Chinese medicine [33]. 

Eating out by these immigrants is mostly done in Chinese restaurants, which are quite different from the Western ones. Its larger tables, round or square, favor sociability and family stay, as a symbol of harmony and union [34]. A wide variety of dishes are served, both hot and cold, maintaining their cooking, such as steam or wok [35], and the attitude of taking typical "Spanish tapas" is not performed. The substitution of olive oil with soybean oil and the scarce use of dairy and fried foods, which are more characteristic of Spanish food, are supported by other authors [32,36,37], and also by the Chinese dietary guidelines [38]. 

Chinese immigrants believe that their food is healthier and, due to this reason, opt to maintain their culinary customs [5]. The health benefits of the traditional Asian diet have been demonstrated in a randomized controlled pilot feasibility study, noting an improvement of insulin sensitivity in Asian Americans [39]. In the present study, ethnic food identity allows Chinese immigrants to enjoy better health, taking into account that the maintenance of eating habits is possible as most Chinese food is now available throughout Europe in native supermarkets, restaurants, and other foreign food markets. 

Traditional eating habits are very common in immigrants, especially among newly arrived immigrants. Benazizi et al. [40] show that adherence to dietary recommendations is more common among immigrants with more than 14 years of residence in Spain. Another study reported dietary changes and health consequences, such as the substantial increase in fat or reduction of carbohydrates and vegetables [41]. Further studies could compare the differences between newly arrived Chinese immigrants in Spain to those who have been there for a longer time.

Chinese immigrants has experienced a notable change in their dietary patterns in recent decades due to three main reasons: acculturation of younger people, weight reduction, and changes in routines due to work. On the one hand, the maintenance of traditional cultural identity among the elderly would explain the higher consumption of high-fat foods and high calorie intake among young people [42]. On the other hand, the food routine is interrupted in many occasions by the demands of the work. Work is the most influential factor responsible for establishing Chinese immigrants’ eating habits. The usual work schedules in Spain differ from those in China. The new situation, characterized by intense work from Monday to Sunday, with working hours of more than 10, causes a complete shift in eating patterns. The promise of bringing money to the family only allows them to think of how to work harder to earn more money, which entails great stress that changes eating habits and produces diseases.

Another important aspect of Chinese food surrounds the relationship between traditional Chinese medicine and food consumption. In TCM, food is conceptualized as having nutritional and functional properties, being used to treat several illnesses [43]. Followers of TCM precepts are commended to eat grains (“grains to nourish” referring to a staple food for maintaining life), fruit (“fruit to assist” referring to a balanced diet), vegetables (“vegetables to supplement” referring to the supplementation of nutrients), and meat (“livestock to benefit” referring to the benefits of these products to the body) [44]. Another concept of TCM is that food has various properties, including warm and hot foods (favorable to the spleen and stomach) and cold and cool foods (detoxifying and laxative) [44]. The benefits of this diet is not totally understood and a recent study found that intakes of sodium, iron, copper, and vitamin E were higher, but fiber, calcium, phosphorus, potassium, selenium, vitamin A, vitamin B1, vitamin B2, and vitamin C were lower than Japanese, American, and Italian diets [45].

### 4.1. Relevance to Clinical Practice

The present study has several clinical implications that should be considered. First, government and public health managers should be aware of the changes of eating habits among Chinese immigrants. These changes may result in important health problems, since they can potentiate the future development of diseases in this population. An example for this assumption is the adoption of a high fat/fried food diet. Second, it is important to be culturally sensitive to the eating patterns of this population, providing alternatives to food not available in Spain as well as making the access to Chinese food easy for this population. Third, these immigrants should receive education concerning working hours and stressful situations in attempt to prevent bad eating habits and improve their quality of life (and healthier diets). Finally, the differences between Chinese and Western/Spanish diets could be a source of stress for this population and should be careful addressed by health professionals. 

Knowing the food culture could infer certain aspects, such as integrating the group into society, and promoting health and well-being. Understanding these intersecting factors could help to ensure culturally appropriate care and optimized health outcomes for Chinese immigrants.

### 4.2. Limitations

This study has some limitations. First, a convenience sample was used, making it difficult to infer that this sample was representative of the Chinese community in Spain. It is important to note that this is a very difficult group to reach out to, due to several reasons, such as lack of time due to work, language, and cultural barriers [46]. In fact, non-representative samples of Chinese immigrants are common in studies assessing Chinese immigrants [47,48] and are explained by the difficulty of assessing this population. Nevertheless, the results are consistent with official data and other studies. Most Chinese immigrants included were from a young middle-age group. Although this could be considered a limitation, the Chinese immigrants in Spain are younger than other immigrants (results are corroborated by previous studies). In addition, as the majority of the sample was from a working population, the main reason for refusal was the lack of time. Although this could be considered a limitation of the study, as described in the Methods section, researchers offered the possibility of answering the survey at another moment of the day if the participant wanted, in an attempt to increase the response rate by this population. Another limitation is the fact that, we did not analyze in depth specific factors influencing eating patterns in Chinese immigrants, as it was not the main objective of this study. Future studies are encouraged to explore the exact reasons for each of these factors in order to expand the interpretation of these findings.

## 5. Conclusions

Chinese immigrants have different eating patterns as compared to the native population in Spain, although these are modified due to the acculturation process and work issues. These findings could help administrators and health professionals in developing strategies to promote health in this population.

## Figures and Tables

**Figure 1 nutrients-13-00766-f001:**
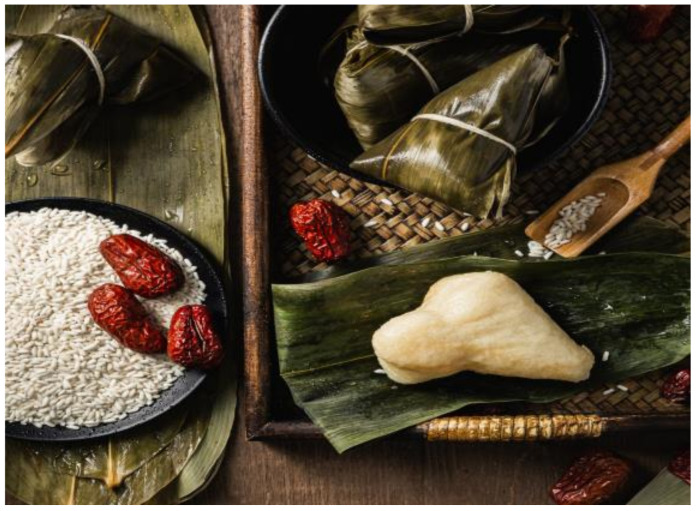
Zongzi. Source: Unplash. Zongzi [24].

**Figure 2 nutrients-13-00766-f002:**
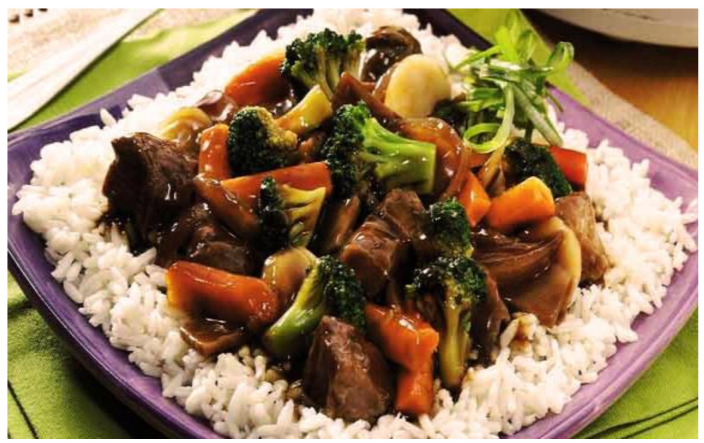
Chop suey. Source: Cocinate el mundo. Chop Suey, los secretos de esta gran receta oriental [25].

**Figure 3 nutrients-13-00766-f003:**
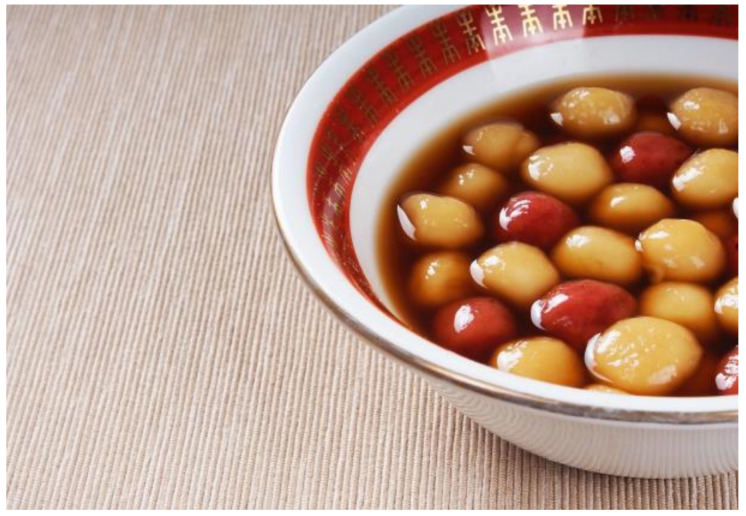
Tangyuan. Source: Unplash. Tangyuan [26].

**Figure 4 nutrients-13-00766-f004:**
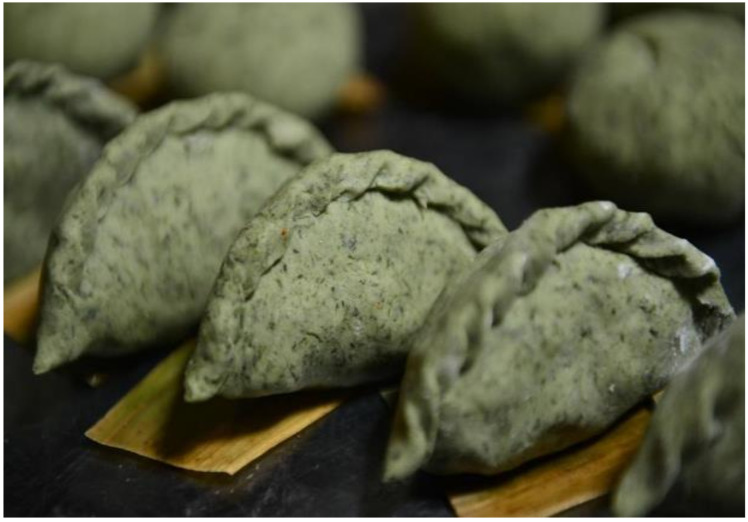
Qingmingguo. Source: Xinhua Español. Qingmingguo, Comida de primavera [27].

**Figure 5 nutrients-13-00766-f005:**
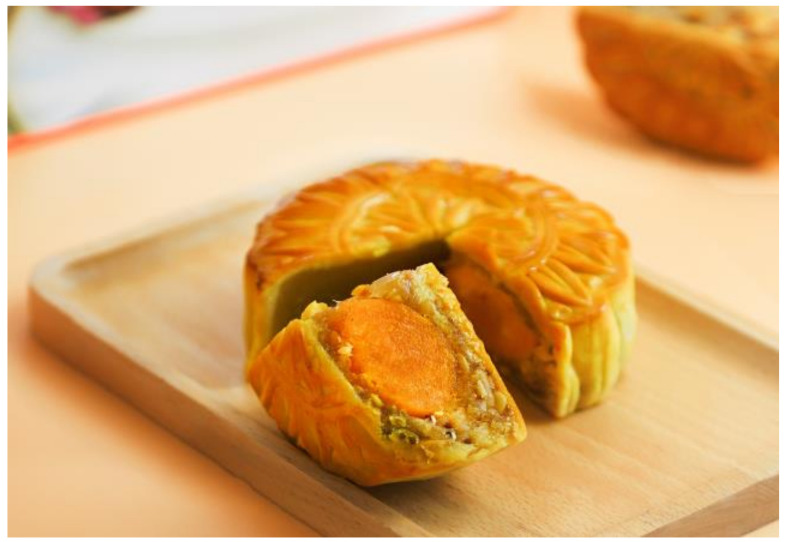
Yuebing. Source: Unplash. Moon cake [28].

**Table 1 nutrients-13-00766-t001:** Sample characteristics.

Variables	Male *M* (*SD*)	Female *M* (*SD*)	Total *M* (*SD*)	Statistics *p* Value
Age (years)	33.1 (7.2)	29.2 (7.4)	30.7 (7.6)	*U* = 1457.5 *p* = 0.003
Years residing in Spain	12.7 (5.7)	10.4 (5.5)	11.3 (5.7)	*U* = 4774.5 *p* = 0.017
	*n* (%)	*n* (%)	*n* (%)	
Sex	51 (38.3)	82 (61.7)	133 (100)	−
Marital status Single Married Living with a partner (not married) Divorced	11 (21.5) 36 (70.6) 3 (5.9) 1 (2.0)	34 (41.5) 46 (56.1) 2 (2.4) 0 (0.0)	45 (33.8) 82 (61.7) 5 (3.8) 1 (0.8)	χ^2^ = 7.35 *p* = 0.062
Level of education Secondary or lower Vocational or university	41 (80.4) 10 (19.6)	62 (75.6) 20 (24.4)	103 (77.4) 30 (22.6)	χ^2^ = 0.41 *p* = 0.521
Employment status Employed Unemployed	50 (98.0) 1 (2.0)	78 (95.1) 4 (4.9)	128 (96.2) 5 (3.8)	χ^2^ = 0.74 *p* = 0.649

**Table 2 nutrients-13-00766-t002:** Emergent categories and distribution of verbatim quotations on food patterns and descriptions of traditional Chinese food.

Theme	Category	Quotations
Food patterns	Differences between Chinese food and Western food	(C-95 woman, 40 years) “We like Spanish food but it’s as if you like Chinese food. You cannot eat Spanish food every day. Your stomach does not allow it”. (C-12 man, 21 years) “A different way used to preserve food is to dehydrate and even salt them”. (C-100 woman, 25 years) “Not much oil is used as in Spain. We eat some churros very similar to those here, but with less oil, much less [It is referred to “Youtiao”, a fried bread sticks widely used for breakfast. Unlike the Spanish churro that is sweet, the Chinese is salty].
	Products and dishes consumedby Chinese immigrants	(C-5 man, 35 years) “The food is totally Chinese from the beginning of the day. We usually have boiled rice or noodle soup for breakfast”. (C-78 woman, 22 years) “We eat vegetables daily and sometimes sauté them with meat in small pieces: chicken, pork, veal”. (C-50 man, 43 years) “Among fish, we like a lot sole or turbot and among the vegetables, we usually use zucchini, onion, and broccoli. The rest are all Chinese vegetables, such as pak choi”. (C-85 man, 37 years) “Chinese people eat many eggs (sometimes eat 2–3 boiled eggs), even for breakfast. When they go on a trip, as they do not have bread in their culture and they cannot make sandwiches, it is normal to eat canned noodles and sometimes they take cooked eggs”. (C-121 woman, 25 years) “The huo guo is very traditional, especially in winter. It is like a pan on fire where we put everything: meat, ball, and vegetables. Everything is boiled; we put sauce on it, sometimes spicy”.
	Modification of eating habits	(C-63 woman, 22 years) “I usually have Spanish breakfast, milk with cereals since I work very early and I don’t have time to prepare an authentic Chinese breakfast. To do this it would need at least an hour [to cook the rice soup and eat it]”. (C-6 man, 39 years) “I can only eat if we take turns at work, because the store cannot be left alone at any time”. (C-128 woman, 32 years) “You can see Chineses eating inside the stores or crouching at the door of the store. They are working and cannot close the business to eat”.
Traditional foods mentioned by the participants		
Figure 1	Zongzi	Triangle shaped dumplings filled as white rice or yellow rice (a form of millet), served dotted with sweetened dates. In Zhejiang province, Jiaxing zongzi is stuffed with marinated ham and wrapped with a leaf.
Figure 2	Chop suey	It is a dish of Chinese–American origin that literally means "mixed pieces." It usually consists of meats (chicken, beef, shrimp, or pork), cooked in a wok with vegetables such as celery, peppers, green beans, among others. It is served with steamed white rice.
Figure 3	Tāngyuán	Tāngyuán are sweet glutinous rice balls, sugar, and almonds. The Chinese New Year celebration ends on the fifteenth lunar day (yuán xiāo jié), known worldwide as the Lantern Festival. Some traditions of this day are to appreciate colorful lanterns through the streets and eat with the family the sweet soup of tāngyuán. Having a round and compact shape, it symbolizes the strong family union. It is also consumed as a dessert at Chinese weddings, and at the Winter Solstice Festival (Dōngzhì).
Figure 4	Qingmingguo	Qingmingguo is a traditional snack eaten on China Tomb Sweeping Day, also known as the Qingming Festival. It is mainly made up of Gnaphalium affine or Jersey Cudweed, a kind of medicinal plant mixed with glutinated rice flour, normally filled with pork, mushrooms, carrots, douya, doufu, and others.
Figure 5	Yuebing	Moon cakes (yuebing) are traditional snacks eaten during the Zhongqiujie or Mid-Autumn Festival, the second largest festival in China. They are filled with soybean red bean paste, lotus seeds with a salted duck egg or nuts. Its roundness represents family reunion, that is, complete happiness and satisfaction.

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
