# Peer review of "Food Patterns among Chinese Immigrants Living in the South of Spain"

_nutrients, 2021, doi:10.3390/nu13030766_

Round 1
Reviewer 1 Report
This paper provides a qualitative description of Chinese immigrants' diet pattern from a sample of 133 immigrants of Chinese origin. The description of their dietary pattern seems to suggest that these immigrants largely follow the Chinese food culture and are not influenced very much by the Spanish food culture.
A few comments:
a. The aim of the study seems to provide inputs on the health strategies targeting these immigrants. Yet, this study does not provide any evidence on the health impact of the Chinese immigrants' dietary choices. Nor does the paper attempt to provide any meaningful discussion.
b. the study does show the differences between Chinese and Western/Spanish diets. However, this difference by itself does not answer the question on why a targeted health strategy is needed. On the contrary, according to the study it seems that the Chinese immigrants believe their own diet is healthier.
c. An important aspect of the study is to explore how the Chinese immigrants modify their eating behavior. Here, the study does not provide much evidence in relation to the characteristics of the immigrants included in the sample. Only some rather casual statements are made (e.g. younger people tend to have different behavior; long work hours lead to skipped meals; etc.).
Some additional issues:
- line 14: "in vulnerable groups, such as immigrants" - is this generally applicable to all ethnical groups?
- line 63-64: "The recent trends in migration shows an increase of 21.9% in Spain’s foreign population over the previous year" - does this mean that within a year, there is a jump of 21.9%?
- line 74-75: rice intake and overweight? I hope the authors would take a look at other studies.
- line 279-280: "The association between the changes on eating habits among Chinese immigrants are important problems to this population..." - not sure what the "association" refers.
- line 297-300: why is a different diet the basis for developing targeted health strategy? (see my general comment b above)
Author Response
Dear reviewer,
Thank you for the revision and for considering our manuscript. Please, find below the responses to each of the comments raised. All changes in this new version of the manuscript are highlighted in yellow.
Reviewer
This paper provides a qualitative description of Chinese immigrants' diet pattern from a sample of 133 immigrants of Chinese origin. The description of their dietary pattern seems to suggest that these immigrants largely follow the Chinese food culture and are not influenced very much by the Spanish food culture.
A few comments:
- The aim of the study seems to provide inputs on the health strategies targeting these immigrants. Yet, this study does not provide any evidence on the health impact of the Chinese immigrants' dietary choices. Nor does the paper attempt to provide any meaningful discussion.
Authors: Thank you for your careful review. We have now modified the Results section in order to highlight how our findings provide evidence on the health impact of the dietary choices of Chinese immigrants. In addition, the discussion section has also been revised and new ideas have been discussed with our findings. We believe the paper is stronger now.
- the study does show the differences between Chinese and Western/Spanish diets. However, this difference by itself does not answer the question on why a targeted health strategy is needed. On the contrary, according to the study it seems that the Chinese immigrants believe their own diet is healthier.
Authors: Thank you for your thoughtful comment. We believe targeted health strategies are needed in order to acknowledge the differences between diets. In other words, the government should be aware of these differences, providing alternatives to food not available in Spain, avoiding unhealthier food patterns in this population, educating immigrants and avoiding bad eating habits (during working hours and stress situations). This could have important repercussions to public health. We agree with you that our “Relevance to Clinical Practice” should be expanded and clarified and we have now modified it according to your comments.
- An important aspect of the study is to explore how the Chinese immigrants modify their eating behavior. Here, the study does not provide much evidence in relation to the characteristics of the immigrants included in the sample. Only some rather casual statements are made (e.g. younger people tend to have different behavior; long work hours lead to skipped meals; etc.).
Authors: The entire manuscript has been reviewed and more detailed information on the characteristics of the sample has been included in the Results section. These characteristics have been linked to the eating patterns of Chinese immigrants and dietary modifications or choices. We have also incorporated quantitative information from the interviews (percentages and statistical significance) in order to provide more evidence in relation to the characteristics of the immigrants included in the sample.
Some additional issues:
- line 14: "in vulnerable groups, such as immigrants" - is this generally applicable to all ethnical groups?
Authors: It has been changed for better understanding.
- line 63-64: "The recent trends in migration shows an increase of 21.9% in Spain’s foreign population over the previous year" - does this mean that within a year, there is a jump of 21.9%?
Authors: It has been removed to avoid confusion. The authors consider that it does not provide relevant information to the reader.
- line 74-75: rice intake and overweight? I hope the authors would take a look at other studies.
Authors: The paragraph on “overweight among Chinese adolescents…” has been modified accordingly.
- line 279-280: "The association between the changes on eating habits among Chinese immigrants are important problems to this population..." - not sure what the "association" refers.
Authors: Thank you for your comment. We agree with you that the sentence was not clear enough. We have now modified it in order to improve readability. We were explaining that the changes on eating habits among Chinese immigrants may result in important health problems and we have now provided an example for this assumption. We have now modified the entire “Relevance to Clinical Practice” section.
- line 297-300: why is a different diet the basis for developing targeted health strategy? (see my general comment b above)
Authors: Thank you for your comment. As previously answered, we have now modified the entire “Relevance to Clinical Practice” in order to make it clear to the reader the importance of targeted health strategies. See if it is appropriate now.
Reviewer 2 Report
By sorting out 133 records of Chinese immigrants' answers to open-ended questions, the authors discuss the dietary patterns of Chinese immigrants living in southern Spain from three aspects: "The differences between Chinese and Western diets", " Products and dishes consumed by Chinese immigrants" and "Modification of eating habits". Finally, the authors draw their main conclusions based on their interview. The idea of this paper is good in general and related to the journal, but I would not recommend it for publication, or at least a majour revision, since there are two severe problems concerning the representativeness of the sample selected for the study and the conclusion based on a descriptive analysis. First, the sample ranged in age from 18 to 55 years old. Only 133 samples chose to participate, and 279 samples refused to participate. As the authors mentioned that the main reason for refusal was lack of time; this is likely to seriously distort the results because of the sample selection. I would suggest authors follow their Table 1 and conduct a statistically analysis on the differences in food patterns among Chinese immigrants. The current version draws their conclusions based on unrepresentative individual interview and claim that there are significant differences, which is not convincing at all. With this regard, I do not see any significant contribution of the study, maybe other reviewer say that. In addition, the authors also need more discussions on their findings. For instance, "Spicy food is consumed by people from Beijing and other areas of Northern China" (lines 182-183). I think this observation is not coincide with the fact in China, probably because of the unreliable sample. Normally, people living in Beijing and Northern China consumed much less spicy food compared to these living in south part of China (e.g., Hunan, Jiangxi, Guizhou, Sichuan, Yunnan, etc.). If this observation is correct, there should be some articles, opinions, or data to support this finding.
Author Response
Dear reviewer,
Thank you for the revision and for considering our manuscript. Please, find below the responses to each of the comments raised. All changes in this new version of the manuscript are highlighted in yellow
Reviewer
By sorting out 133 records of Chinese immigrants' answers to open-ended questions, the authors discuss the dietary patterns of Chinese immigrants living in southern Spain from three aspects: "The differences between Chinese and Western diets", “Products and dishes consumed by Chinese immigrants" and "Modification of eating habits". Finally, the authors draw their main conclusions based on their interview. The idea of this paper is good in general and related to the journal, but I would not recommend it for publication, or at least a majour revision, since there are two severe problems concerning the representativeness of the sample selected for the study and the conclusion based on a descriptive analysis. First, the sample ranged in age from 18 to 55 years old. Only 133 samples chose to participate, and 279 samples refused to participate.
Authors: Thank you for your review and for considering our manuscript. We have now addressed all your suggestions and concerns in order to improve the clarity of our study. In order to provide additional information for your first comment (the sample), which we consider very important, we decided to meticulously explain our reasons for choosing this sample size. Please see below.
First, we agree with you that our sample does not represent the entire Chinese immigrant population in Spain, and for this reason, we have incorporated this information as a limitation in the manuscript. It is important to note that this is a very difficult group to reach out, due to several reasons such as lack of time due to work, language and cultural barriers, reasons also shared by other authors assessing this population (Hernando, Sabidó, Ronda, Ortiz & Casabona; 2014)[1]. In fact, non-representative samples of Chinese immigrants are found in other studies (Lee & Brann, 2012[2]; Lee, Suchday & Wylie., 2012[3]; Ribas, 2013[4]; Vargas, 2014[5]), and are explained by the difficulty of assessing this population.
In our study, as described in the manuscript, we worked very hard to include the greatest number of immigrants as possible. This could be seen in the sample size of 133 participants, which for qualitative studies is considered a very good sample size. In our study, using the current best practices for qualitative studies, sample size was determined using the concepts of "saturation" and "information power". In order to make generalizability even better, we decided to include a greater number of participants than needed by qualitative guidelines.
In order to illustrate the appropriateness of our sample, we decided to compare our sample with other studies published in the journal “Nutrients” as shown below:
- Turner et al., 2021. Can Monitoring Make It Happen? An Assessment of How Reporting, Monitoring, and Evaluation Can Support Local Wellness Policy Implementation in US Schools: interviews with 39 superintendents.
- Dao et al., 2021. Cultural Influences on the Regulation of Energy Intake and Obesity: A Qualitative Study Comparing Food Customs and Attitudes to Eating in Adults from France and the United States: 25 adults in France with or without overweight/obesity participated in semi-structured interviews.
- Højer et al., 2020. Play with Your Food and Cook It! Tactile Play with Fish as a Way of Promoting Acceptance of Fish in 11- to 13-Year-Old Children in a School Setting—A Qualitative Study: The design was a qualitative exploratory multiple-case design using participant observation in a school setting. Six classes were recruited from the Eastern part of Denmark (n = 132).
Concerning the sample range in age from 18 to 55 years old, according to data from the National Institute of Statistics (2016-2019), the average age of the foreign population in Spain has ranged between 32.09 and 32.14 years, which shows that the immigrant population is young.
Regarding Chinese immigrants, the following table extracted from the National Institute of Statistics shows that the ages of the Chinese immigrants in our study coincide with those of the entire Chinese population in Spain. This was considered for our study, and therefore, we decided to clarify this information in the manuscript. Please, see Methods and Limitations section.
As the authors mentioned that the main reason for refusal was lack of time; this is likely to seriously distort the results because of the sample selection.
Authors: Thank you for raising this question. As explained previously, this is a common problem while assessing Chinese immigrants and this was included as a limitation. However, the researchers offered the possibility of answering the survey in other moment if the participant wants and this may have reduced the refusals. We have now included this information in the manuscript.
I would suggest authors follow their Table 1 and conduct a statistically analysis on the differences in food patterns among Chinese immigrants. The current version draws their conclusions based on unrepresentative individual interview and claim that there are significant differences, which is not convincing at all. With this regard, I do not see any significant contribution of the study, maybe other reviewer say that.
Authors: We appreciate your comment. Although the present study is designed to be a qualitative study, we have included some quantitative information as well. We agree with you that showing this data could provide more information to those readers more familiar to quantitative studies. Therefore, we have now included quantitative data concerning the sample characteristics in Table 1 and concerning the food patterns throughout the text (showing the significant differences in the text).
In addition, the authors also need more discussions on their findings. For instance, "Spicy food is consumed by people from Beijing and other areas of Northern China" (lines 182-183). I think this observation is not coincide with the fact in China, probably because of the unreliable sample. Normally, people living in Beijing and Northern China consumed much less spicy food compared to these living in south part of China (e.g., Hunan, Jiangxi, Guizhou, Sichuan, Yunnan, etc.). If this observation is correct, there should be some articles, opinions, or data to support this finding.
Authors: Thank you for your comment. In our study, participants underscored that persons from Northern China consumed more spicy food than those from the Southern part of China. Before including this information in the manuscript, we have checked if such information was accurate and, indeed, it is. We have now included some references supporting our data, including a recent paper published concerning the dietary preferences and diabetic risk in China. This was a large nationwide study, which corroborates with our findings (See Zhao et al. J Diabetes. 2020 Apr; 12(4):270-278). We have now included it and other references to support this information.
[1] Hernando, C., Sabidó, M., Ronda, E., Ortiz Barreda, G., & Casabona, J. (2014). Una revisión sistemática de estudios longitudinales de cohorte sobre la salud en poblaciones migradas. Medicina Social, 8(2), 81-94.
[2] Lee, A., & Brann, L. (2015). Influence of Cultural Beliefs on Infant Feeding, Postpartum and Childcare Practices among Chinese-American Mothers in New York City. J Community Health, 40, 476–483.
[3] Lee ,YS, Suchday, S., & Wylie Rosett, J. (2012). Perceived social support, coping styles, and Chinese immigrants' cardiovascular responses to stress. Int.J. Behav. Med., 19, 174–185.
[4] Ribas da Costa, MA. (2013). Satisfação dos imigrantes chineses com os Cuidados de Saúde Primários: relatório de um estudo realizado numa unidade de saúde da ARSC. Curso de Mestrado em Enfermagem Comunitaria. Escola Superior de Enfermagem de Coimbra.
[5] Vargas Urpi, M. (2014). Actitudes y percepciones del colectivo chino en cuanto a la comunicación en los servicios públicos: ejemplos del contexto catalán. Lengua y migración 6(1), 5-41.
Reviewer 3 Report
- This manuscript has 30% similarity, the material and methods section is copy paste from other publications (see report attached), please modify it, authors can paraphrase
- Authors must revise the English writing
- Use Asian instead of Asiatic
- First person (we, our, us, etc) must not be used in formal reports, please modify. First person may only be used for citations with quotation marks when providing examples of the participants’ comments
- Have the authors considered for further studies comparing immigrants who recently arrived to Spain (maybe <2 years) with those with longer time (~10 years)? It would be interesting to see how their perception changes

Author Response
Dear reviewer,
Thank you for the revision and for considering our manuscript. Please, find below the responses to each of the comments raised. All changes in this new version of the manuscript are highlighted in yellow
Reviewer
This manuscript has 30% similarity, the material and methods section is copy paste from other publications (see report attached), please modify it, authors can paraphrase.
Authors: We sincerely apologize for this inconvenience. The “Material and methods” section has been modified in order to remove this similarity. In addition, we have verified the rest of the document using the Turnitin anti-plagiarism program.
Authors must revise the English writing.
Authors: The final translation was carried out and proofread by Pangeanic, SL. a professional translation company that offers fast and reliable aI-enhanced human translation. As a translation company, Pangeanic provides specialized technical translation and specialized translation services with a team of in-house translators and freelance expert linguists (for more information visit http://pangeanic-translations.us )
Use Asian instead of Asiatic.
Authors: We agree with your comment and we have now modified accordingly.
First person (we, our, us, etc) must not be used in formal reports, please modify. First person may only be used for citations with quotation marks when providing examples of the participants’ comments.
Authors: It has been modified as suggested.
Have the authors considered for further studies comparing immigrants who recently arrived to Spain (maybe <2 years) with those with longer time (~10 years)? It would be interesting to see how their perception changes
Authors: We agree with your comment. It has been included in the discussion section as suggested.
Reviewer 4 Report
In the submission, there are no grammatical and stylistic errors. However, the part of Introduction section is tedious and hard to follow. These description are occupied too much spaces; these authors should be written more concisely.
Author Response
Dear reviewer,
Thank you for the revision and for considering our manuscript. Please, find below the responses to each of the comments raised. All changes in this new version of the manuscript are highlighted in yellow
Reviewer
In the submission, there are no grammatical and stylistic errors. However, the part of Introduction section is tedious and hard to follow. These description are occupied too much spaces; these authors should be written more concisely.
Authors: Thank you for your review. The Introduction section has now been rewritten for better understanding in a concise manner.
In addition to this “Response to reviewers”, the authors have considered a document with figures to be included as a new file. These figures represent "Traditional foods mentioned by the participants" in Table 2. The idea with this new figure is to allow readers to get an understanding of the characteristics of these traditional foods. Using a graphical approach will help readers to be familiar with the aspects of these traditional foods.
Round 2
Reviewer 2 Report
N.A